# Preparation and Performance of a Grid-Based PCL/TPU@MWCNTs Nanofiber Membrane for Pressure Sensor

**DOI:** 10.3390/s25103201

**Published:** 2025-05-19

**Authors:** Ping Zhu, Qian Lan

**Affiliations:** School Instrument and Electronics, North University of China, Taiyuan 030051, China; lanqian0901@163.com

**Keywords:** pressure sensor, nanofiber-based, wider dynamic range, the sensitivity, response speed

## Abstract

The intrinsic trade-off among sensitivity, response speed, and measurement range continues to hinder the wider adoption of flexible pressure sensors in areas such as medical diagnostics and gesture recognition. In this work, we propose a grid-structured polycaprolactone/thermoplastic-polyurethane nanofiber pressure sensor decorated with multi-walled carbon nanotubes (PCL/TPU@MWCNTs). By introducing a gradient grid membrane, the strain distribution and reconstruction of the conductive network can be modulated, thereby alleviating the conflict between sensitivity, response speed, and operating range. First, static mechanical simulations were performed to compare the mechanical responses of planar and grid membranes, confirming that the grid architecture offers superior sensitivity. Next, PCL/TPU@MWCNT nanofiber membranes were fabricated via coaxial electrospinning followed by vacuum-filtration and assembled into three-layer planar and grid piezoresistive pressure sensors. Their sensing characteristics were evaluated by simple index-finger motions and slide the mouse wheel identified. Within 0–34 kPa, the sensitivities of the planar and grid sensors reached 1.80 kPa^−1^ and 2.24 kPa^−1^, respectively; in the 35–75 kPa range, they were 1.03 kPa^−1^ and 1.27 kPa^−1^. The rise/decay times of the output signals were 10.53 ms/11.20 ms for the planar sensor and 9.17 ms/9.65 ms for the grid sensor. Both sensors successfully distinguished active index-finger bending at 0–0.5 Hz. The dynamic range of the grid sensor during the extension motion of the index finger is 105 dB and, during the scrolling mouse motion, is 55 dB, affording higher measurement stability and a broader operating window, fully meeting the requirements for high-precision hand-motion recognition.

## 1. Introduction

Flexible pressure sensors have achieved remarkable research progress in diverse fields such as medical diagnostics, human–machine interaction, robotics control, and gesture recognition in daily life. Compared to traditional rigid pressure sensors, their superior performance is primarily reflected in high-sensitivity, ultra-wide detection range, rapid response, low detection limits, and exceptional flexibility and durability. These breakthroughs stem from the synergistic integration of advancements in materials science, microstructural design, and manufacturing processes. Specifically, innovations in nanomaterials (e.g., graphene and MXene), hierarchical architectures (e.g., pyramid arrays and porous networks), and scalable fabrication techniques (e.g., 3D printing and electrospinning) collectively enable unprecedented performance metrics, such as sub-Pascal detection limits, millisecond-level response times, and stable operation under extreme mechanical deformation. This convergence of interdisciplinary innovations positions flexible pressure sensors as transformative tools for next-generation wearable electronics, precision healthcare monitoring, and adaptive robotic systems [1,2,3,4]. Representative flexible substrates—including poly-ε-caprolactone (PCL), thermoplastic polyurethane (TPU), and polydimethylsiloxane (PDMS)—coupled with nanoconductive fillers such as graphene and carbon nanotubes [5,6,7,8,9] have propelled sensor technology from the paradigm of rigid integration toward flexible, conformal architectures. Breakthroughs along this trajectory are manifested in three key dimensions: (i) enhanced compatibility with biological tissues, (ii) robustness under extreme environmental conditions, and (iii) pushing the performance envelope of sensing devices.

However, the high sensitivity of flexible pressure sensors typically relies on drastic reconstruction of the conductive network (e.g., interlayer sliding of graphene sheets). Their response speed is governed by how quickly the real contact area within that network changes and how stable those contacts remain, while a broad working range requires the material to preserve its structural integrity under large loads. Current technologies struggle to reconcile these demands, leading to three classic trade-offs: (1) sensitivity vs. measurement range. Maintaining high sensitivity usually comes at the expense of a wide pressure window. For example, graphene micro-array sensors exhibit diminished sensitivity at both very low and very high pressures, restricting their usable range [9,10]. MXene/PDMS composites can reach sensitivities of 0.061 kPa^−1^ in the range of 39–171 kPa [11]. Theoretical models further show that, once compressive strain exceeds ~50%, the density of conductive pathways saturates, causing sensitivity to plummet [12,13]. (2) Sensitivity vs. response speed. Introducing micro/nano texturing enhances sensitivity but also complicates charge-transport pathways, slowing the sensor’s electrical response [14]. (3) Measurement range vs. response speed. Achieving a broad range demands highly elastic materials, yet such materials recover slowly from deformation, again limiting response speed [15]. Consequently, the intrinsic conflict among sensitivity, response speed, and operating range remains the principal bottleneck in the advancement of flexible pressure sensors.

To address these challenges, structural innovation has emerged as a decisive lever. Conventional planar membranes (e.g., flat PDMS films) suffer from uniform stress fields that concentrate at the center easily triggering nonlinear deformation [16]. In contrast, mesh-type membranes—such as electrospun nanofibre networks—realise hierarchical stress -dispersion through a three-dimensional interpenetrating framework (with an overlap-junction density of ~3.6 × 10^4^ mm^−2^). Under load, their real contact area can increase by nearly one third, boosting sensitivity relative to planar films; yet their improvement is still limited when an ultrabroad pressure range is required [17]. Building on this insight, we propose a graded grid-pattern membrane architecture. Poly-ε-caprolactone (PCL, biodegradation period 3–24 months) [18] and thermoplastic polyurethane (TPU, service temperature –40 °C to 120 °C) nanofibres are intertwined to form a molecular interpenetrating network, while multi-walled carbon nanotubes (MWCNTs) provide a microscopic three-dimensional conductive pathway [19]. Regularly arrayed square apertures are cut into the nanofibre mat, partitioning the global strain of the original planar film into wall bending and node stretching, thereby imparting a macroscopic gradient. ANSYS 2021 mechanical simulations are employed to elucidate how the grid structure amplifies sensitivity. Subsequently, PCL/TPU@MWCNT nanofibre sensors are fabricated and evaluated in terms of mechanical response, sensitivity, and simple index-finger motion detection. The results demonstrate a synergistic optimisation of sensitivity, response speed, and dynamic range, offering a new strategy to overcome the long-standing “high-sensitivity–fast-response–wide-range” dilemma.

## 2. Mechanical Simulation of the Nanofibre Membrane

### 2.1. Model Construction and Parameter Settings

Finite-element simulations were carried out in ANSYS 2021 Workbench, using the SpaceClaim module to build both planar and grid-patterned nanofibre-membrane models. Static analyses were then performed to compare their stress-field characteristics and sensitivity trends under a uniformly distributed load. A representative membrane measuring 30 mm × 26 mm was selected as the computational domain.

(1) Grid membrane. Sixteen square apertures (each 5 mm × 4 mm) were introduced, separated by 2 mm to minimise spurious interactions during calculation.

(2) Thickness. The membrane thickness was fixed at 0.05 mm.

These geometrical parameters strike a balance between capturing the microstructural features of the nanofibre network and ensuring that the finite-element model can faithfully predict macroscopic behaviour under various load conditions. The resulting CAD models are illustrated schematically in Figure 1.

First, the material parameters of each constituent were defined. Because poly-ε-caprolactone (PCL), thermoplastic polyurethane (TPU), and multi-walled carbon nanotubes (MWCNTs) differ greatly in their intrinsic properties, a mass-fraction weighted-average method was adopted to estimate the effective properties of the composite; the addition of MWCNTs markedly enhances both mechanical strength and electrical conductivity. Based on the intrinsic properties of PCL, TPU, and MWCNTs and their mixing ratios, the effective material constants used in the finite-element model were assigned, as summarised in Table 1.

After the material constants were assigned, the models were meshed with an element size of 0.5 mm. One face of each membrane was designated as the supporting surface and treated as a fixed support, i.e., all nodes on that face were fully constrained to eliminate displacement. This allows the opposite face to deform freely. A uniform load was then applied to the free surface to replicate the operational stresses experienced by the nanofibre membranes. Because the pressure generated by an adult index finger during bending varies widely—ca. 9–75 kPa during everyday activities such as writing or pinching [20]—a pressure of 75 kPa was selected for the present simulation.

### 2.2. Stress Distribution Mechanism and Sensitivity Performance Analysis

Figure 2 illustrates the strain distributions of the models under the 75 kPa load.

As shown in Figure 2a,b, a rectangular planar nanofibre membrane subjected to a uniform load displays a biaxially symmetric stress field dominated by the long edges; strain is highly concentrated at the corners and mid-sections of the long sides, while only minor deformation occurs in the central region. This steep strain gradient leads to elastic-modulus mismatch failure when the pressure exceeds about 5 kPa; the conductive network can no longer reconstruct, the relative resistance change (ΔR/R) saturates, and the sensor’s sensitivity therefore declines sharply under higher loads [21].

By contrast, Figure 2c,d reveals a markedly different stress landscape for the grid-patterned membrane; the local strain around each aperture far exceeds the global average and the geometric discontinuities along the hole edges give rise to pronounced stress concentrations. According to Equation (1), substituting the local stress values for the grid (1.2936 × 10^−5^) and planar (6.5977 × 10^−6^) membranes shows that the sensitivity of the grid nanofibre sensor is roughly twice that of its planar counterpart. The rectangular perforations partition the overall deformation into (i) wall bending and (ii) node stretching, enabling a transition from global load bearing to local energy dissipation and thereby introducing a unique, multilevel stress-redistribution mechanism.

The sensitivity is defined by:(1)$$S=\frac{\pi \cdot \sigma_{max}}{E}\times \frac{1}{h}$$
where *S* is the sensitivity of the sensor, *σ_max_* is the maximum stress in the membrane, *E* is the Young’s modulus of the material, *π* is the piezoresistive coefficient of the membrane material, and *h* is the membrane thickness.

## 3. Fabrication and Measurement of the PCL/TPU@MWCNTs Nanofiber-Based Pressure Sensor

### 3.1. Preparation of PCL/TPU@MWCNTs Nanofiber Membrane

#### 3.1.1. Dispersion of MWCNT

Multi-walled carbon nanotubes (MWCNTs) were first dispersed by a combined stirring–ultrasonication protocol: (1) preparation of the suspension. A measured amount of MWCNT powder was added to deionised water to obtain a concentration of 2.0 g L^−1^. Sodium dodecyl sulphate (SDS) was introduced as a surfactant to aid dispersion. (2) Mechanical pre-dispersion. The mixture was agitated with a laboratory stirrer to break up large agglomerates. (3) Ultrasonic treatment. The pre-dispersed slurry was transferred to an ultrasonic cleaner and sonicated for 2 h. This step yielded a homogeneous, stable MWCNT suspension suitable for subsequent processing.

#### 3.1.2. Fabrication of the PCL/TPU@MWCNT Nanofibre Membrane and Assembly of the Pressure Sensor

The materials used in this experiment are listed in Table 2, and the instruments are shown in Table 3.

Preparation of the PCL/TPU spinning solutions: (1) PCL solution—the required amount of PCL pellets was weighed with an analytical balance and dissolved in a dichloromethane (DCM)/N,N-dimethylformamide (DMF) mixed solvent (mass ratio 7:3) to obtain a 12 wt% PCL solution. (2) TPU solution—TPU powder was weighed and dissolved in pure DMF to form a 10 wt% TPU solution. (3) Homogenisation—both solutions were mechanically stirred for 5 h at room temperature until complete dissolution produced clear, uniform spinning dopes.

Coaxial electrospinning of the composite membrane: (1) the PCL and TPU solutions were loaded into two separate 5 mL syringes and mounted on the syringe pumps of a coaxial electrospinning setup. (2) They were electrospun at 28 kV with a tip-to-collector distance of 20 cm to produce the PCL/TPU nanofibre mat. (3) The as-spun mat was cut into circular discs 50 mm in diameter.

In situ deposition of MWCNTs: (1) 7.5 mL of the previously prepared MWCNT suspension was vacuum-filtered onto each PCL/TPU disc. (2) It was rinsed repeatedly with deionised water; then, the coated membrane was dried in a thermostatic oven for 2 h to yield the PCL/TPU@MWCNT nanofibre composite.

In flexible pressure sensors, the key role of nylon mesh as a dielectric regulating layer and mechanical support is due to its unique mesh structure that regulates the physical properties of the dielectric layer. The specific mechanism can be divided into the following three aspects [22]: (1) dynamic regulation of porosity and optimization of electric field distribution. The nonuniform distribution of pores can induce stress concentration effects and improve the detection sensitivity of small pressure signals through local strain amplification mechanisms. For example, when pressure is applied, the deformation amplitude of the pore edge area in the mesh structure is much greater than that in the dense area, forming a gradient dielectric constant field. (2) Topology reconstruction of conductive pathways. When nylon mesh is used as a supporting skeleton, its grid cells can serve as dispersed templates for conductive fillers such as carbon nanotubes or silver nanowires. When pressure causes grid deformation, the number of contact points and path density of the conductive network undergo dynamic reconstruction, resulting in nonlinear changes in resistivity. (3) Inhibition of mechanical hysteresis effect. The high elastic modulus of nylon mesh can limit the plastic deformation of the dielectric layer material and reduce the structural relaxation phenomenon during repeated loading through elastic recovery force.

Therefore, sensor construction was conducted as follows: (1) the composite membrane was trimmed to 30 mm × 26 mm rectangles. Using a surgical blade, 16 square apertures (5 mm × 4 mm each) were cut in a regular array across the membrane surface. (2) A 20 mm × 20 mm nylon mesh was placed on top as a dielectric-tuning and mechanical-support layer; this also enhanced flexibility. The effective overlapping area between mesh and membrane was 20 mm × 20 mm. (3) The stack was encapsulated with 55 µm thick polyimide tape to provide electrical insulation and protect the device from ambient moisture and air.

A schematic of the membrane fabrication and sensor assembly process is presented in Figure 3.

### 3.2. Testing Methods

The sensitivity tests were carried out on a custom platform comprising a digital force–displacement tester and a multimeter, as illustrated in Figure 4.

Mechanical loading: the sensor was placed on the platen of the force tester. A compressive load was applied over a 10 mm × 10 mm area at the centre of the device while the cross-head displacement was recorded.

Electrical read-out: the two output leads of the sensor were connected to the multimeter to monitor the instantaneous resistance *R*.

Sensitivity definition: the sensitivity data in this paper were obtained by averaging five experiments and selecting the value closest to the mean as the sensitivity. Sensitivity S is defined as the slope of the curve relating the relative resistance change to the applied pressure [21]:(2)$$S=\frac{\left(\Delta R/R_{0}\right)\times 100\%}{\Delta P}$$
where *R*_0_ is the sensor’s initial resistance without pressure, Δ*R* is the change in resistance under applied pressure, and Δ*P* is the change in applied pressure with the initial pressure set to 0.

The sensor’s response and recovery times were characterised with a constant-voltage source and an oscilloscope. The device was affixed to a finger joint, and its leads were connected to a 5 V constant-voltage supply. The oscilloscope time base was set to the millisecond scale to capture waveform changes as the finger bent and straightened. Ten rise-and-fall events were recorded; the rise and decay times were then averaged to obtain the sensor’s dynamic response metrics.

## 4. Test Results

### 4.1. SEM and XRD of Membranes

Figure 5 depicts the morphology of PCL/TPU@MWCNTs membranes. The membranes prepared with 5 mL filtration volumes, where MWCNTs bonded to fibres via physical entanglement or chemical grafting, forming ordered, loose three-dimensional conductive networks along fibre axes. This structure benefits from the effective dispersion treatment of MWCNTs and provides reinforcement without excessively restricting the deformation of the polymer matrix.

X-ray diffraction (XRD) analysis is used to gain a deeper understanding of the crystal structure of nanofibre membranes. The analytical results are shown in Figure 6, where an extremely strong diffraction peak appears at 2θ = 26.2°, corresponding to the (210) crystal plane of the PCL as well as the (002) crystal plane of the MWCNTs. Two smaller diffraction peaks appear at 2θ of 42.2° and 43.6°, corresponding to the (100) crystal plane of MWCNTs [23,24]. In addition, there are no other obvious diffraction peaks on the diffractograms, which indicates that there are no other impure crystal impurities on the surface of the fibre membrane. The peak at the (002) crystallographic plane is much higher than that at the (100) crystallographic plane, which indicates that the MWCNTs have higher surface energy at the (002) side, which can induce faster growth of MWCNTs. The grain sizes of MWCNTs on the (002) and (100) facet side were calculated to be about 28.4 nm and 29.7 nm according to Scherrer’s formula. This size is similar to the field emission scanning electron microscopy characterization map.

### 4.2. Mechanical Response Analysis

A touchscreen-controlled tensile tester, paired with a high-precision multimeter, was used to track the resistance of the PCL/TPU@MWCNT nanofibre pressure sensors while their length was gradually extended. After an initial static strain of 8%, the resistance was logged; the strain was then raised stepwise to 12%, 16%, 20%, 24%, 28%, and 32%, holding each level for 600 s before recording the new resistance value (Figure 7). Without load, the planar sensor exhibited a resistance of 368 Ω, whereas the grid sensor read 247 Ω. Once the desired tensile load was applied and held, the relative resistance at each step remained stable, showing virtually no drift or fluctuation. In both sensors, the resistance change increased linearly with strain and the response curves stayed smooth throughout the experiment, free from decay or oscillation. At 32% strain, the grid sensor’s relative resistance had grown by 76.3%, representing a 19.3% higher gain than that of the planar sensor. This pronounced improvement confirms that incorporating the grid membrane markedly boosts sensitivity, most likely by enhancing the reconfiguration of the internal conductive network.

### 4.3. Sensitivity Analysis

Specifically, grid sensors with aperture sizes of 4 mm × 3 mm, 5 mm × 4 mm, and 6 mm × 5 mm were added and their sensitivity was measured. Within the range of 0–34 kPa, the sensitivities of 4 mm × 3 mm, 5 mm × 4 mm, and 6 mm × 5 mm are 2.07 kPa^−1^, 2.24 kPa^−1^, and 2.15 kPa^−1^, respectively. Within the range of 35–75 kPa, the sensitivities are 1.12 kPa^−1^, 1.27 kPa^−1^, and 1.20 kPa^−1^, respectively. The grid sensor with an aperture of 5 mm × 4 mm has good sensitivity. The sensitivity curve is shown in Figure 8.

The sensitivities of the two PCL/TPU@MWCNT nanofibre sensors were measured, and the results are plotted in Figure 9. Within the 0–34 kPa range, the planar and grid sensor exhibited sensitivities of 1.80 kPa^−1^ (standard deviation 0.033 kPa^−1^) and 2.24 kPa^−1^ (standard deviation 0.020 kPa^−1^), respectively; in the 35–75 kPa interval, the corresponding values were 1.03 kPa^−1^ (standard deviation 0.017 kPa^−1^) and 1.27 kPa^−1^ (standard deviation 0.047 kPa^−1^). In both pressure windows, the linearity was excellent, with R^2^ > 0.99. Relative to the planar PCL/TPU@MWCNT sensor, the optimised grid sensor achieved a 24.44% increase in sensitivity at low pressure and a 23.3% increase at high pressure. Enhanced sensitivity allows the device to register smaller pressure fluctuations with greater precision. According to Equation (1), the sensitivity of the grid sensor was obtained as 2.44 kPa^−1^ based on finite element prediction, and the actual experimental measurement yielded a sensitivity of 2.24 kPa^−1^ with a relative error of 8.93%. The small discrepancy between finite-element predictions and experimental data confirms that the simulations accurately capture the sensor’s mechanical response under varying loads, providing reliable guidance for future design iterations.

### 4.4. Output-Response Analysis

Instantaneous pressure stimuli were applied to the sensors by repeatedly bending and straightening a finger, and the rise and fall times of their electrical responses were recorded (Figure 10).

Planar sensor: Figure 10a shows a rise time of ≈10.53 ms: the relative resistance (Δ*R*/*R*_0_) jumps rapidly to ≈3.4% and then stabilises. Figure 10b gives a fall time of ≈11.20 ms, i.e., the interval from load removal to recovery of the baseline. After unloading, Δ*R*/*R*_0_ quickly returns to nearly 0%, demonstrating prompt restoration of the conductive network. The difference between rise and fall times is about 0.67 ms, which likely reflects the finite time required for the network to rebuild as the film rebounds from a very rapid deformation.

Grid-pattern sensor: in Figure 10c, the rise time is shortened to ≈9.17 ms and Δ*R*/*R*_0_ reaches ≈4.2%, remaining essentially constant for the next 86 ms. Figure 10d shows a fall time of ≈9.65 ms.

Compared with the planar device, the grid sensor thus displays faster rise and recovery dynamics, in agreement with the improvements observed in its static response.

### 4.5. Index Finger Simple Motion Analysis

PCL/TPU@MWCNT nanofibre pressure sensors were used to test the movement changes of the index finger when bending and tapping the mouse. To test the movement changes of the index finger when bending, both the planar and grid sensor were affixed to the finger joint, and an oscilloscope recorded their output voltages while the finger was smoothly bent from 0° to 45°. The experimental setup is depicted in Figure 11. After envelope detection of the collected time-domain signals by the planar sensor and the grid sensor, an FFT was performed to obtain envelope amplitude–frequency curves, as shown in Figure 12. Figure 12a–c are plots of the time-domain signal, envelope, and logarithmic amplitude spectrogram collected by the planar sensor, and Figure 12d–f are for a grid sensor.

In Figure 12c,f, the energy of active finger motion is concentrated in the 0–0.5 Hz band; according to Ref. [19], this corresponds to smooth finger movement in the low-frequency (0–2 Hz) range.

The planar sensor—Figure 12c shows that the spectral magnitude decreases fairly uniformly within 0–0.5 Hz, providing a dynamic range of 0–60 dB.

The grid sensor—in Figure 12f, the spectrum falls much more steeply over the same band, yielding a dynamic range of 0–105 dB.

A steeper drop at low frequencies indicates that the measurement system responds more rapidly to low-frequency inputs and reaches its steady state sooner, thereby enhancing steady-state accuracy. Consequently, the grid sensor exhibits better stability and a far wider dynamic range in the low-frequency region, making it fully capable of high-precision recognition of hand motions such as grasping and extension.

Based on the above analysis, we selected a grid sensor to test the movement changes of the index finger when gripping the mouse tightly. Figure 13 illustrates the response curve of a grid sensor for mouse wheel operation. Figure 13a is a plot of the measurement site, Figure 13b is a time-domain signal, and Figure 13c is a logarithmic amplitude spectrogram. It can be seen from the time-domain waveforms diagram that the sensor senses the location source of the force and the magnitude of the force and has good durability and short response time. As can be seen from the logarithmic amplitude spectrogram, the signal spectral lines are concentrated in the 0–0.5 Hz band and are capable of providing a dynamic range of 55 dB.

### 4.6. Cycling Stability of the Grid Sensor

To test the cycling performance of the grid sensor, we have increased the length of the finger bending/straightening experiment to measure the cycling stability of the sensor.

Figure 14 illustrates that the experimental time of finger bending/straightening was extended to 1200 s, and one bending/straightening cycle was completed every 2 s on average. The dynamic cyclic stability curve of the sensor in 1200 s was obtained. As can be seen from the Figure 14 the relative resistance change basically maintains the fluctuation in the interval of 4.2–4.5% during the test time of 1200 s. This indicates that, for the sensor in the dynamic cyclic loading process, its resistance change has a certain degree of regularity and repeatability and there is no substantial deviation or sudden change, reflecting that the sensor has a better dynamic cyclic stability.

## 5. Conclusions

By introducing a graded grid-pattern membrane, the PCL/TPU@MWCNT nanofibre pressure sensor achieves a synergistic optimisation of sensitivity, response speed, and operating range. In the 0–34 kPa window, the planar and grid devices display sensitivities of 1.80 kPa^−1^ and 2.24 kPa^−1^, respectively; in the 35–75 kPa range, the values are 1.03 kPa^−1^ and 1.27 kPa^−1^. Their rise/decay times are 10.53 ms/11.20 ms (planar) and 9.17 ms/9.65 ms (grid). Both sensors reliably detect active index-finger motions in the 0–0.5 Hz band, with dynamic ranges of ≈60 dB for the planar device and ≈105 dB for the grid device. Finally, the mechanical process of the index finger rolling mouse pulley is measured by using a grid sensor, and the dynamic response in the range of 0–0.5 Hz is 55 dB. During the 1200 s testing time, the relative resistance change of the grid sensor remained fluctuating within the range of 4.2–4.5%. Owing to its higher sensitivity, faster electrical response, much broader dynamic range, and the cycling stability, the grid sensor is fully capable of high-precision recognition of hand actions such as grasping and extension.

## Figures and Tables

**Figure 1 sensors-25-03201-f001:**
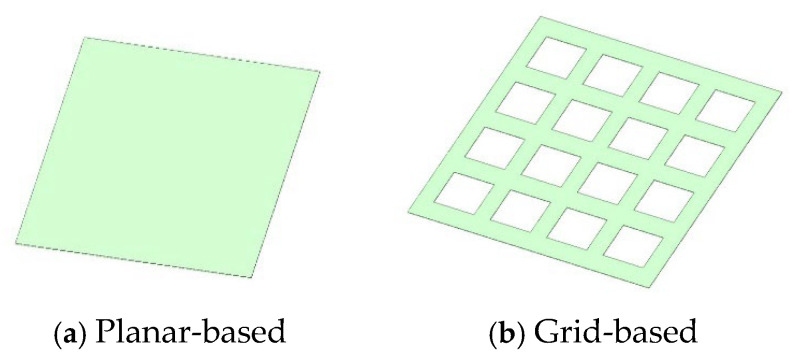
Geometrical models of membranes.

**Figure 2 sensors-25-03201-f002:**
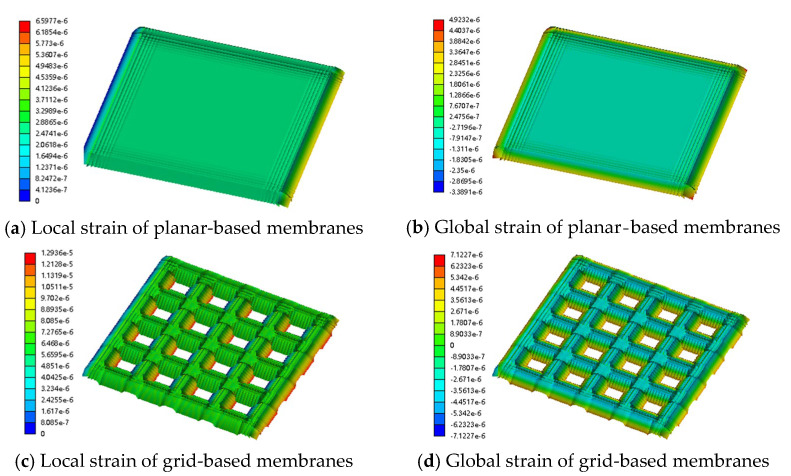
Stress–strain contour map of planar and grid-based membranes.

**Figure 3 sensors-25-03201-f003:**
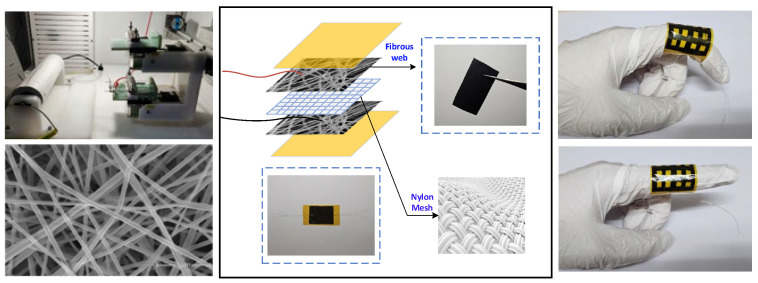
Schematic diagram of electrospinning device and sensor assembly.

**Figure 4 sensors-25-03201-f004:**
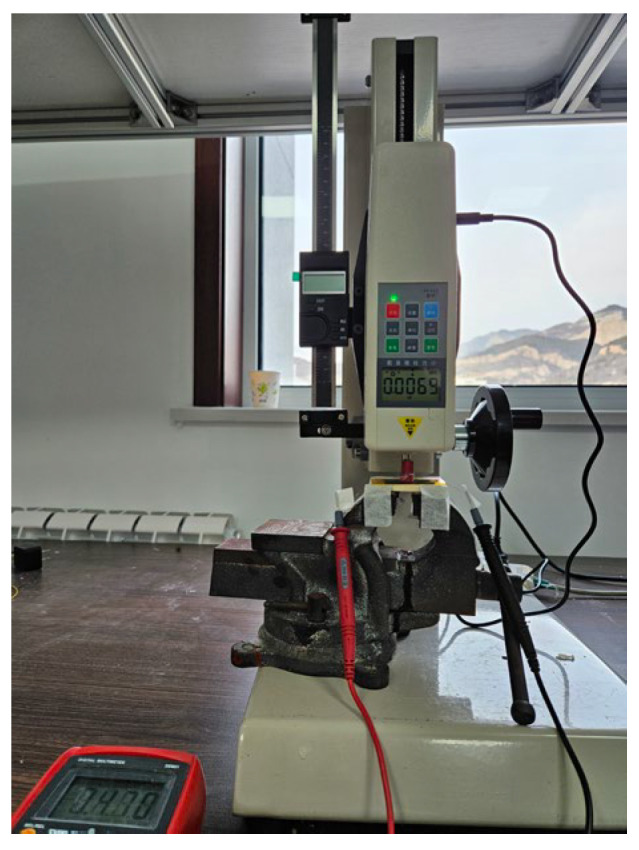
Sensitivity test platform.

**Figure 5 sensors-25-03201-f005:**
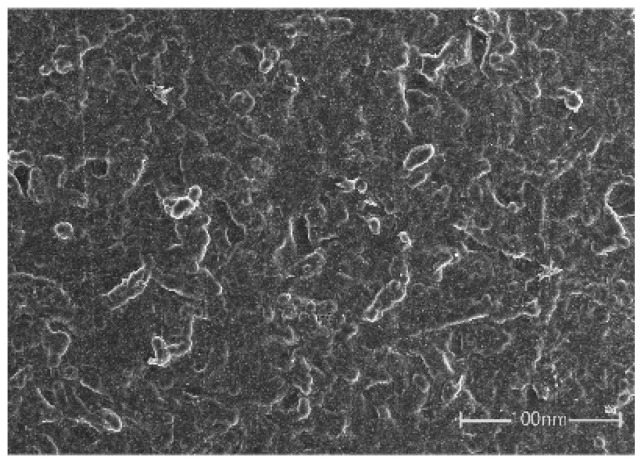
SEM of PCL/TPU@MWCNTs membranes.

**Figure 6 sensors-25-03201-f006:**
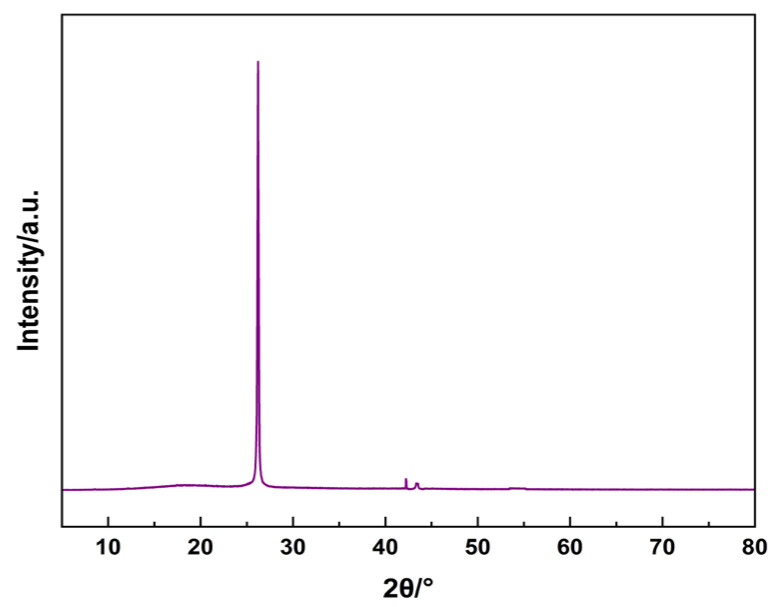
XRD patterns of Membrane.

**Figure 7 sensors-25-03201-f007:**
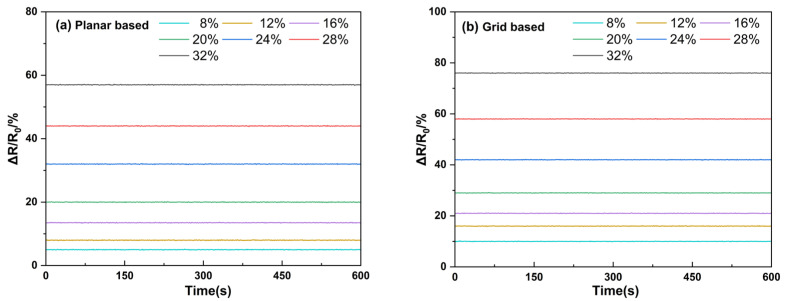
Resistance change curve of the sensor under stretching conditions.

**Figure 8 sensors-25-03201-f008:**
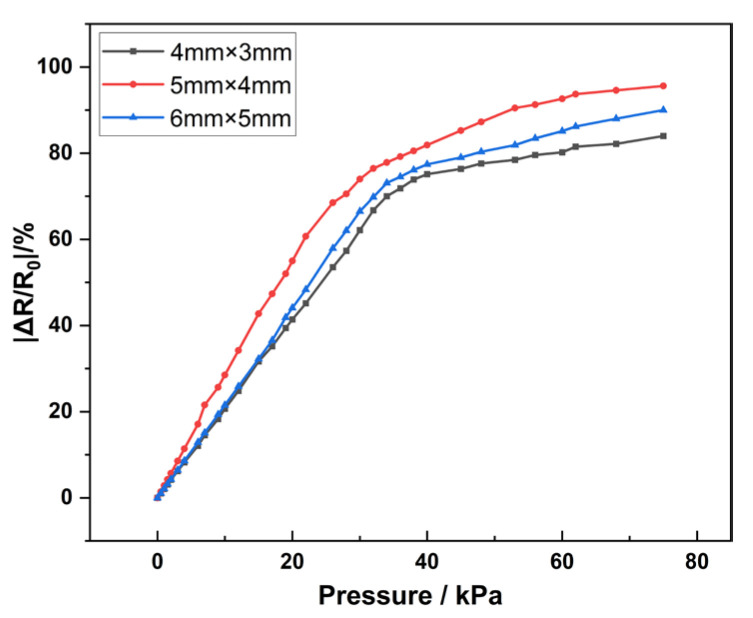
Sensitivity of the grid sensor.

**Figure 9 sensors-25-03201-f009:**
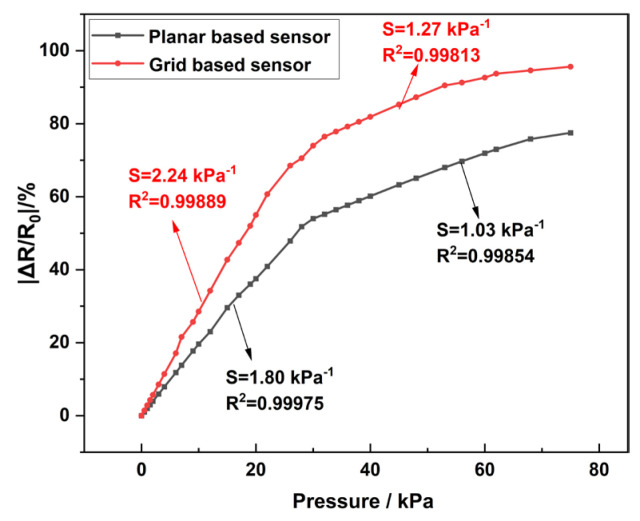
Sensitivity curve of the sensor.

**Figure 10 sensors-25-03201-f010:**
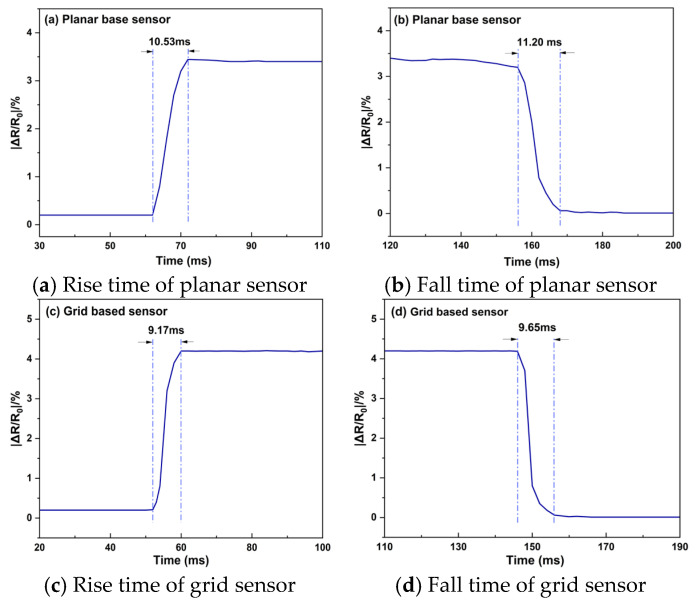
Output-response characteristics of the sensor.

**Figure 11 sensors-25-03201-f011:**
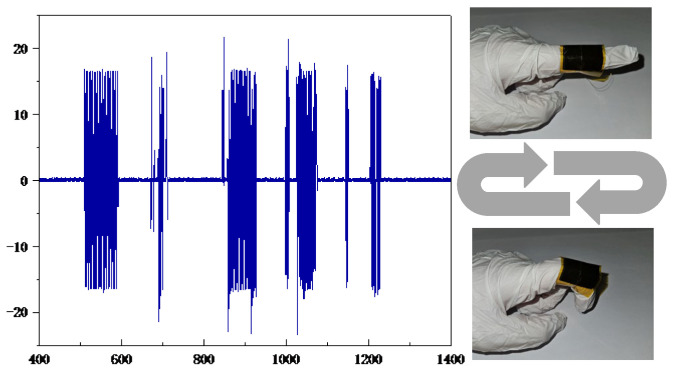
Simple motion measurement of the index finger.

**Figure 12 sensors-25-03201-f012:**
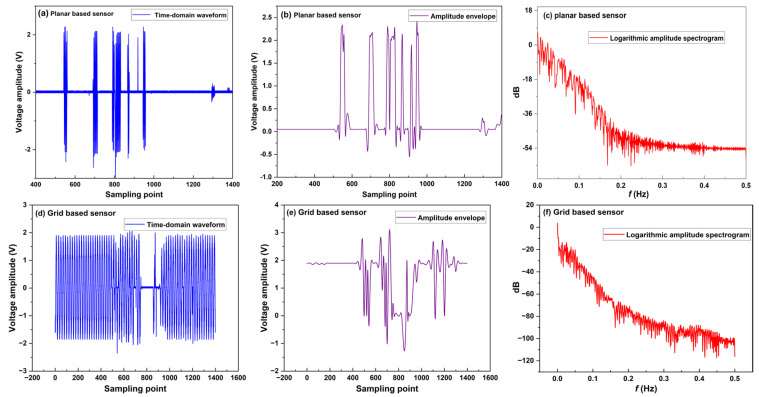
The motion measurement data when the index finger stretches and contracts. (**a**) The time-domain signal of planar sensor; (**b**) envelope of planar sensor; (**c**) logarithmic amplitude spectrogram of planar sensor; (**d**) the time-domain signal of grid sensor; (**e**) envelope of grid sensor; (**f**) logarithmic amplitude spectrogram of grid sensor.

**Figure 13 sensors-25-03201-f013:**
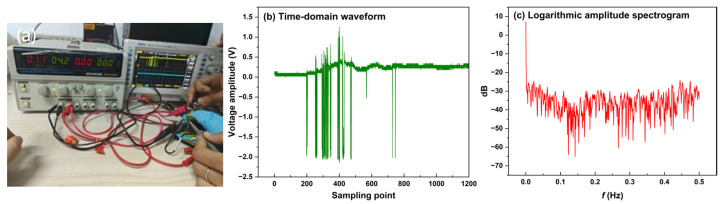
The measurement data when the index finger is scrolling. (**a**) The measurement site; (**b**) the time-domain signal; (**c**) the logarithmic amplitude spectrogram.

**Figure 14 sensors-25-03201-f014:**
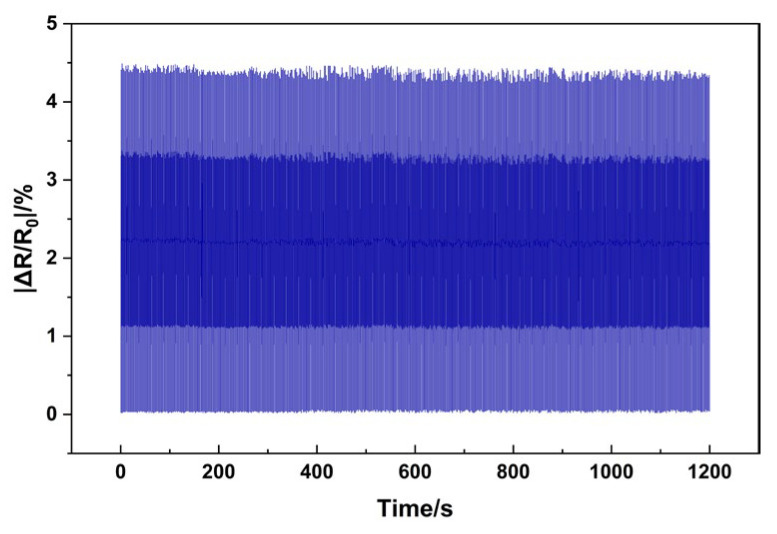
The cycling stability of the sensor.

**Table 1 sensors-25-03201-t001:** Material parameters of nanofibre membrane.

	Elastic Modulus GPa	Poisson’s Ratio	Thermal Conductivity Coefficient W/m·K	Density g/cm^3^
PCL/TPU@MWCNTs	1	0.35	0.7	1.15

**Table 2 sensors-25-03201-t002:** Experimental material.

Material Name	Specification	Manufacturer
Polycaprolactone (PCL)	6800	Swedish Bersto Specialty Chemicals Company
Thermoplastic Polyurethane (TPU)	1195a	BASF AG, Germany
Multi walled carbon Nanotubes (MWCNTs)	>98%	Merck Darmstadt, Germany
Sodium Dodecyl Sulfonate	Chemically pure	Tianjin Kemio Chemical Reagent Co., Ltd., China
Dichloromethane (DCM)	Chemically pure	Chengdu Cologne Chemical Co., Ltd., China
Dimethylformamide (DMF)	Chemically pure	Chengdu Cologne Chemical Co., Ltd., China

**Table 3 sensors-25-03201-t003:** Experimental instruments.

Equipment Name	Model	Manufacturer
Functional electrospinning equipment	NS-1	Qingdao Junada Technology Co., Ltd., China
Analytical balance	JJ124BC	Changshu Shuangjie Testing Instrument Factory, China
Four way magnetic heating stirrer	CJJ-4	Changzhou Guowang Instrument Equipment Co., Ltd., China
Ultrasonic cleaning machine	JP020	Shenzhen Jiemeng Technology Co., Ltd., China
Vacuum filtration machine	R410-MF31	Xiamen Loken Instrument Co., Ltd., China
SEM	Apreo 2C	Thermo Fisher Scientific, USA
Touch screen measurement and control tensile strength tester	JSM-7900F	Guangdong Zhongye Jingke Instrument Equipment Co., Ltd., China

## Data Availability

All the data are included in the article.

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
