# Peer review of "Preparation and Performance of a Grid-Based PCL/TPU@MWCNTs Nanofiber Membrane for Pressure Sensor"

_sensors, 2025, doi:10.3390/s25103201_

Round 1
Reviewer 1 Report
Comments and Suggestions for Authors
This manuscript presents a flexible pressure sensor's design, fabrication, and characterization based on a grid-structured PCL/TPU@MWCNTs nanofiber membrane. The authors aim to address the trade-offs between sensitivity, response speed, and range in flexible sensors by employing a gradient grid structure. While the work presents an interesting approach and demonstrates promising results, several points require clarification and revision before the manuscript can be considered for publication:
1. The authors state that the difference between finite element simulation predictions and experimental results was minimal. However, the manuscript lacks quantitative data or a detailed error analysis to support this claim. Please provide a more direct comparison to validate the simulation model against the experimental findings.
2. The dispersion quality of MWCNTs is critical for the electrical and mechanical properties of the resulting composite nanofiber membrane. Although the use of stirring and ultrasonication for dispersion is mentioned, no evidence (e.g., SEM/TEM images, particle size analysis) is provided to confirm the homogeneity and effectiveness of the MWCNT dispersion within the PCL/TPU matrix. Please include characterization data to demonstrate adequate dispersion.
3. The manuscript identifies the nylon mesh as a dielectric adjustment layer and mechanical support. However, the mechanism by which it "adjusts" the dielectric properties is not explained. Furthermore, its specific contribution to the overall mechanical response (e.g., flexibility, strain distribution) and the sensing performance (e.g., influence on resistance change, baseline resistance) requires more detailed discussion and potentially supporting data.
4. There appears to be an error in the caption for Figure 2. Based on the images, (b) seems to represent the planar membrane.
5. The experimental equipment needs to be mentioned in the article, including the name and manufacturer of the test equipment, the sources of PCL, TPU, MWCNTs, etc.
6. The formatting of the reference list needs careful checking and correction according to the journal's guidelines. Some symbols before the page number are ":", while others are "," .
Author Response
Please refer to the attached response file.

Reviewer 2 Report
Comments and Suggestions for Authors
This manuscript designs a grid-structured PCL/TPU@MWCNTs nanofiber-based pressure sensor. By using gradient grid structure design to adjust strain distribution and reconstruct conductive network, the sensitivity, response speed, and range have been improved. However, this manuscript still needs further improvement in terms of data analysis depth and some detailed arguments. Therefore, this manuscript may be acceptable for publication after major revision. The comments to the authors are given as below:
- The proposal of gradient grid structure effectively solves the contradiction between sensitivity and range, but further discussion is needed on the specific implementation of "gradient", such as the gradient changes in grid size or density, which should be reflected in experiments.
- The performance comparison data between planar and grid sensors (such as sensitivity and response time) lacks statistical analysis (such as error range or repeatability testing). It is suggested to supplement the standard deviation of multiple experiments.
- The finger motion test (0-0.5 Hz) validated the practicality of the sensor, but did not involve complex gesture or multi-point pressure detection scenarios. It is suggested to increase tests in a wider range of application scenarios.
Author Response

(The authors gave the same response as above.)

Reviewer 3 Report
Comments and Suggestions for Authors
In this manuscript, the authors reported a grid-based PCL/TPU@MWCNTs nanofiber membrane, and demonstrated their performance as pressure sensor. This work seems to be useful for this field. However, the following problems should be addressed before further consideration of publication:
- The Introduction should be revised to clearly show the innovation. The performance of the pressure sensor especially the superiority should be stated when compared with other researches.
- The clarity of many images in the manuscript is low, such as Figure 3 and Figure 4. The details of the setup and samples are not easy to read.
- All the figures need to be revised with consistent layout and style to improve the readability. The captions for each subfigures should also be added.
- Material characterization of the PCL/TPU@MWCNTs nanofiber membrane should be added including SEM, EDS, Raman spectra, etc.
- Performance tests (index finger motion) of the pressure sensor seems to be quite simple. More effective tests and appropriate application scenarios should be added for better demonstration.
- The cycling performance of the sensor should be tested. Besides, the depth could be improved if the authors provided insights of sensing mechanisms and properties of the samples.
- The references should be updated considering some novel researches in recent years. Typical examples of flexible sensors should be included: 10.1007/s12274-024-6806-z, 10.1002/EXP.20210033.
Author Response

(The authors gave the same response as above.)

Round 2
Reviewer 1 Report
Comments and Suggestions for Authors
The authors have revised the manuscript as requested, and I think now it can be published without revision.
Reviewer 2 Report
Comments and Suggestions for Authors
Author has made a detailed response to the comments. The revised manuscript has been greatly improved. I recommend publication of the manuscript.
Reviewer 3 Report
Comments and Suggestions for Authors
The revisions have been checked.